# Quantification of aminobutyric acids and their clinical applications as biomarkers for osteoporosis

Zhiying Wang [1,8], Liangqiao Bian[2,8], Chenglin Mo [1], Hui Shen [3], Lan Juan Zhao [3], Kuan-Jui Su [3], Maciej Kukula[2], Jauh Tzuoh Lee[4], Daniel W. Armstrong[5], Robert Recker[6], Joan Lappe[6], Lynda F. Bonewald [7], Hong-Wen Deng [3] & Marco Brotto [1]*

Osteoporosis is a highly prevalent chronic aging-related disease that frequently is only detected after fracture. We hypothesized that aminobutyric acids could serve as biomarkers for osteoporosis. We developed a quick, accurate, and sensitive screening method for aminobutyric acid isomers and enantiomers yielding correlations with bone mineral density (BMD) and osteoporotic fracture. In serum, γ-aminobutyric acid (GABA) and (R)-3-aminoisobutyric acid (D-BAIBA) have positive associations with physical activity in young lean women. D-BAIBA positively associated with hip BMD in older individuals without osteoporosis/osteopenia. Lower levels of GABA were observed in 60–80 year old women with osteoporotic fractures. Single nucleotide polymorphisms in seven genes related to these metabolites associated with BMD and osteoporosis. In peripheral blood monocytes, dihydropyrimidine dehydrogenase, an enzyme essential to D-BAIBA generation, exhibited positive association with physical activity and hip BMD. Along with their signaling roles, BAIBA and GABA might serve as biomarkers for diagnosis and treatments of osteoporosis.

[1] Bone-Muscle Research Center, College of Nursing and Health Innovation, University of Texas at Arlington, Arlington, TX 76019, USA. [2] Shimadzu Center for Advanced Analytical Chemistry, University of Texas at Arlington, Arlington, TX 76019, USA. [3] Tulane Center of Bioinformatics and Genomics, Department of Biostatistics and Data Science, Tulane University, New Orleans, LA 70112, USA. [4] AZYP LLC - Separation & Analytics, Arlington, TX 76019, USA. [5] Department of Chemistry and Biochemistry, College of Science, University of Texas at Arlington, Arlington, TX 76019, USA. [6] School of Medicine Osteoporosis Research Center, Creighton University, Omaha, NE 68122, USA. [7] Indiana Center for Musculoskeletal Health, School of Medicine, Indiana University, Indianapolis, IN 46202, USA. [8] These authors contributed equally: Zhiying Wang, Liangqiao Bian. *email: marco.brotto@uta.edu

Osteoporosis, one of the most prevalent chronic aging-related bone diseases, often goes undetected until the first fragility fracture occurs, causing patient suffering, high healthcare costs and societal burden. Aminobutyric acids are a physiologically relevant class of nonproteinogenic amino acids due to their production from different organs and their potential roles in health and disease. Aminobutyric acids comprise three type of isomers: α-aminobutyric acid (AABA), β-aminobutyric acid (BABA), and γ-aminobutyric acid (GABA). Moreover, pairs of mirror-image isomers with the same molecular weights and related structures, also known as enantiomers, including (R)- and (S)-2-aminobutyric acid (D- and L-AABA), (R)- and (S)-3-aminobutyric acid (D- and L-BABA), and (R)- and (S)-3-aminoisobutyric acid (D- and L- BAIBA), have been identified in this class of molecules. These enantiomers have identical physical and chemical properties in anisotropic environments, but can have different biological actovotoes.

AABA has been investigated as a general biomarker of various conditions such as alcoholic liver injury, sepsis, malnutrition, and multiple organ failure[1–3]. Elevated plasma AABA is also linked to tuberculosis and many pediatric metabolic diseases like Reye's syndrome, etc[3]. BABA is a natural product of plants controlled by the plant's immune system important for the treatment of various plant diseases[4]; however, there are no associations between BABA and human diseases. GABA is a major inhibitory neurotransmitter in the central nervous system that regulates communication between neurons, and it is implicated in behavior, cognition, oxidative stress, glucose tolerance, and pathological conditions such as Alzheimer's disease and diabetes[5,6]. Studies in MCK-PGC1α (muscle-specific overexpression of peroxisome proliferator-activated receptor gamma coactivator 1-α) mice demonstrated that GABA levels in both plasma and skeletal muscles significantly increased following exercise. GABA may function as a PGC1α-mediated myokine and play an important role in the release of growth hormone in adaptive response to exercise[7].

L-BAIBA is produced from valine through a mitochodrial enzyme 4-aminobutyrate aminotransferase (ABAT)[8]. D-BAIBA is produced in the cytosol of liver and kidney from thymine in a metabolic pathway involving dihydropyrimidine dehydrogenase (DPYD), dihydropyrimidinase (DPYS), and β-ureidopropionase (UPB1)[9,10]. Some amount of L-BAIBA and D-BAIBA can convert to each other via the stereo-isomerization reaction between their metabolites, L-methylmalonate semialdehyde (L-MMS) and D-methylmalonate semialdehyde (D-MMS), through enzymatic or nonenzymatic mechanisms[11]. But to date this conversion process has not been described clearly. A previous study reported that BAIBA attenuates insulin resistance, suppresses inflammation, and induces fatty acid oxidation via the AMP-activated protein kinase (AMPK) and peroxisome proliferator-activated receptor δ (PPARδ) signaling pathway in skeletal muscle[12]. BAIBA levels increase in circulation in response to exercise, thus it serves as a contraction-induced myokine that increases energy expenditure and participates in exercise-induced protection from metabolic diseases like type 2 diabetes[13–15]. However, the distribution of L- and D- BAIBA is contradictory as most studies report total BAIBA[9].

A major limitation of the aforementioned studies is that none of these molecules were completely separated and directly quantified, but relied on derivatizations and deconvolutions. The importance of the effects of stereoselectivity in biological functions cannot be underscored. Mirror-image enantiomers can exhibit significanlty different biological potencies, dose–response relationship, and toxicity. For example, R-thalidomide has been reported to be responsible for sedative effects, but its S-enantiomer and related derivatives were linked to teratogenic effects[16]. During the period of 2010–2014, 81 of 127 new molecular entities (NMEs) approved by the Food and Drug Administration (FDA) in the United States were chiral[17]. Moreover, 77 single enantiomers were the major component among these 81 chiral NMEs[17]. This stereoselectivity was also observed among the isomeric aminobutyric acids. In comparison with D-BAIBA, L-BAIBA was 100–1000 times more potent in protecting osteocytes from reactive oxygen species (ROS)-induced cell death, was shown to signal through the Mas-Related G Protein-Coupled Receptor Type D (MRGPRD), was secreted by contracted muscle from C57Bl/6 mice, and maintained osteocyte viability and reduced both bone and muscle loss due to hindlimb unloading of mice[18]. Thus, in mice the L-enantiomer of BAIBA might act as the enantiomer secreted by muscle that activates MRGPRD signaling in bone in mice. Therefore, identification and quantification of isomeric aminobutyric acids including all single enantiomers in circulation/tissues is necessary to elucidate the molecular mechanisms underlying their biological activities, and may provide a better understanding in the diagnosis and potential treatments for the related diseases.

In comparison with other techniques, liquid chromatography–mass spectrometry (LC-MS) is more selective and sensitive, and therefore, has been widely used as a major analytical approach for bioanalysis. But the current quantification methods for aminotyric acid isomers/enantiomers using LC-MS are typically compromised with unresolved peak separations, large sample volumes, time-consuming derivatization, and long analysis times[19,20]. An LC-MS/MS method, reported by Vemula et al. (2017), is the only published method to separate eight isomeric aminobutyric acids simultaneously[21]. However, their derivatization process and LC-MS analysis was time and labor intensive and, more importantly, five out of the eight aminobutyric acid isomers/enantiomers could not be chromatographically separated. It was therefore important to develop a fast, simple, and sensitive LC-MS method to chromatographically separate and thus accurately quantify these isomeric compounds in biological samples to more precisely determine their pathphysiological roles.

We present a LC-MS/MS method that enables baseline separation with sensitive detection of six underivatized aminobutyric acid isomers (D,L-BAIBA, GABA, D,L-AABA, and L-BABA) in minimal amounts of biological fluid samples (10 μL) with a short LC-MS analysis (27 min), specifically for application to the study of osteoporosis. We report specific signatures and correlations between serum GABA or D-BAIBA levels and BMD and physical activities in female cohorts of different ages, low or high BMD values, and with or without osteoporotic fractures (OF). The applicability of this method may lead to significant advances in the detection and treatment of osteoporosis.

## Results

**Development and validation of LC-MS/MS method.** For the first time, to our knowledge, we successfully implemented a methodology for complete separation of six isobaric aminobutyric acids using liquid chromatography without derivatization (Fig. 1a, Supplementary Table 1 and Supplementary Fig. 1). It was then validated by determining a suitable surrogate matrix, calibration linearity, limit of detection, accuracy, and precision. The validation process followed the FDA's Bioanalytical Method Development Guidance for Industry[22].

Linear regression lines were constructed for each biological matrix versus 5% bovine serum albumin (BSA) in phosphate-buffered saline (PBS, pH 7.4) at 11 concentrations, and linear coefficient values were calculated as presented in Supplementary Table 2. A slope of $K = 1$ from the linear equation implies correlation and thus demonstrates similar matrix effects exerted

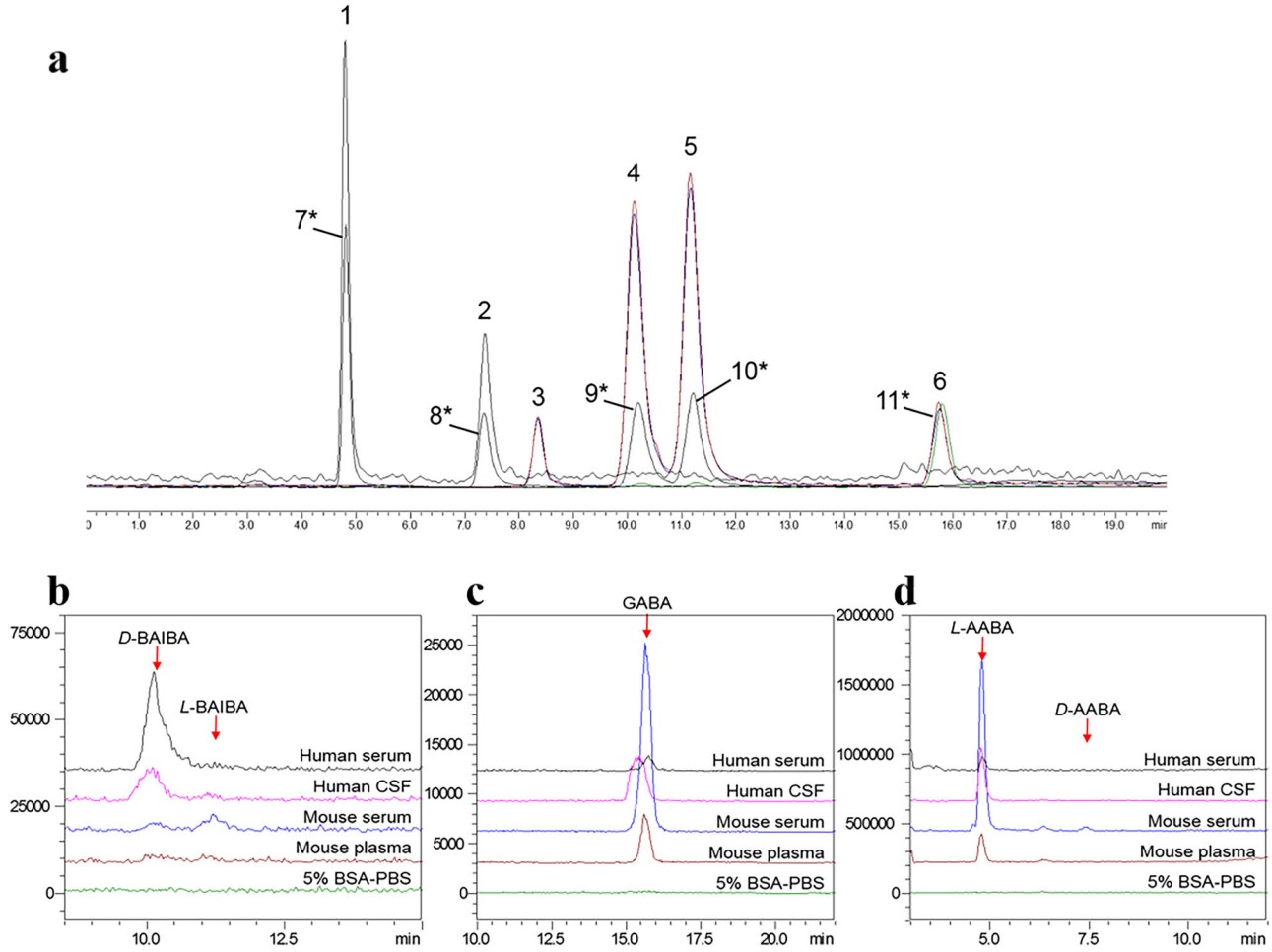

**Fig. 1 Chromatograms of each isomeric aminobutyric acid in different matrices. a** Representative MS/MS chromatogram of isomeric aminobutyric acids and deuterated standards in methanol. Peaks # labelled correspond to the compounds 1–11 in Supplementary Table 1. **b–d** Chromatograms of BAIBA (**b**), GABA (**c**), and AABA (**d**) in surrogate matrix (5% BSA-PBS) and blank human serum, human CSF, mouse serum and mouse plasma.

by these two matrices[23–26]. As shown in Supplementary Table 2, the slopes of the linear equations from four sets of solutions are close to unity (0.87–1.12) for these analytes, indicating parallelism of matrix effects presented between the surrogate matrix selected and biological fluids. Comparison of chromatograms in 5% BSA-PBS and various physical fluid samples in Fig. 1b–d also demonstrated that no biological interference existed in BSA-PBS for the detection of BAIBA, GABA, or AABA.

We conducted calibration curves for five aminobutyric acid isomers of interest in the ranges of 0.02–20.48 μM for GABA, L-BAIBA and D-BAIBA, and 0.16–163.84 μM for L-AABA and D-AABA. The calibration ranges varied depending on the sensitivity of the individual analyte and its expected endogenous level in biological fluid samples. The calibration curves exhibited excellent linearity ($R^2 = 0.9992$–0.9998) in the calibration ranges established for all five aminobutyric acids (Table 1).

The limits of detection (LODs) were 0.01 μM for GABA, L-BAIBA and D-BAIBA, and 0.08 μM for L-AABA and D-AABA (Table 1). These LOD values correspond to the detection limits of 10 pg and 80 pg, respectively, for the two groups of aminobutyric acids. As shown at the lowest concentrations of each calibration range in Table 1, the limit of quantification (LOQs) were 0.02 μM for GABA, L-BAIBA and D-BAIBA, and 0.16 μM for L-AABA and D-AABA. As shown in Table 1, the accuracies for all three levels of quality control (QC) samples are in the range of 90.3–110.8% for all five aminobutyric

acid isomers analyzed. Relative standard deviation (RSD, an indication of precision) for all three levels of QC samples were in the range of 2.2–9.2%.

**Quantification of aminobutyric acids in biological fluids**. To demonstrate the feasibility of the present LC–MS/MS method, we measured aminobutyric acids in pooled plasma, serum and cerebrospinal fluid (CSF) from healthy human subjects with different ethnicity as well as from mice of different ages and sex. The results summarized in Table 2 indicate that the predominant type of BAIBA in humans is the R-enantiomer (D-BAIBA), with concentrations of 0.34 ± 0.02 μM and 0.14 ± 0.01 μM in human serum and CSF. In contrast, L-BAIBA was the primary isomeric form in mouse (0.05 ± 0.01 μM). D-BAIBA was also detected in the pooled healthy mouse serum, but the concentration was lower than the LOQ and thus no quantification was obtained (Fig. 1b). This finding with pooled human sera was further confirmed by individual human serum samples. D-BAIBA serum levels were remarkably higher than L-BAIBA in all six human serum samples with various ethnicity, age, and gender (Table 2). The large L-AABA peak at about 4.8 min was observed in all fluid samples from both humans and mice, but D-AABA was not detected in the samples (Fig. 1d). This suggests that L- is the only AABA enantiomer naturally existing in biological fluids analyzed in the present study.

**Table 1 Validation results of limit of detection (LOD), calibration range, linearity, precision, and accuracy of the new assay.**

| | Calibration curve | | | Method Validation | | | |
|---|---|---|---|---|---|---|---|
| | LOD (µM) | Linear range (µM) | Linearity ($R^2$) | Prepared concentration (µM) | Measured concentration (µM) | Precision (CV %) | Accuracy (%) |
| L-BAIBA | 0.01 | 0.02–20.48 | 0.9998 | 0.32 | 0.33 ± 0.02 | 6.5 | 101.6 |
| | | | | 1.28 | 1.32 ± 0.08 | 5.8 | 103.5 |
| | | | | 5.12 | 5.22 ± 0.20 | 3.9 | 102.0 |
| D-BAIBA | 0.01 | 0.02–20.48 | 0.9992 | 0.32 | 0.29 ± 0.03 | 9.2 | 90.3 |
| | | | | 1.28 | 1.42 ± 0.07 | 5.2 | 110.8 |
| | | | | 5.12 | 5.45 ± 0.24 | 4.4 | 106.5 |
| GABA | 0.01 | 0.02–20.48 | 0.9999 | 0.16 | 0.17 ± 0.01 | 6.0 | 104.4 |
| | | | | 0.64 | 0.70 ± 0.03 | 3.8 | 109.1 |
| | | | | 2.56 | 2.58 ± 0.07 | 2.7 | 101.0 |
| L-AABA | 0.08 | 0.16–163.84 | 0.9994 | 2.56 | 2.38 ± 0.09 | 3.7 | 93.1 |
| | | | | 10.24 | 10.70 ± 0.26 | 2.5 | 104.5 |
| | | | | 40.96 | 42.28 ± 1.91 | 4.5 | 103.2 |
| D-AABA | 0.08 | 0.16–163.84 | 0.9995 | 2.56 | 2.52 ± 0.05 | 2.2 | 98.3 |
| | | | | 10.24 | 11.00 ± 0.37 | 3.4 | 107.5 |
| | | | | 40.96 | 41.35 ± 1.62 | 3.9 | 100.9 |

Eight-point calibration curves were prepared to the following concentrations: 0.02, 0.04, 0.08, 0.16, 0.64, 2.56, 10.24, and 20.48 µM for L-BAIBA and D-BAIBA; 0.02, 0.04, 0.08, 0.32, 1.28, 5.12, 10.24, and 20.48 µM for GABA; 0.16, 0.32, 0.64, 1.28, 5.12, 20.48, 81.92, and 163.84 µM for L-AABA and D-AABA. Mean ± SD ($n = 5$) for method validation

**Applications of developed method in clinical studies**. To investigate the relationship between isomeric aminobutyric acids and osteoporosis, serum samples from two clinical studies were analyzed using the newly developed method: (1) 54 older women (age: 48–80 years) with or without single or multiple fractures (Supplementary Table 3); and (2) 136 younger women (age 21–41 years) with low or high BMD values (hip) (Supplementary Table 4). In the older women group, the concentrations of three quantified aminobutyric acids in non-osteoporotic (Control) vs. osteoporotic (CASE) groups were 0.71–2.24 µM vs. 0.38–1.97 µM, D-BAIBA, 0.09–0.20 µM vs. 0.08–0.16 µM, GABA, and 2.36–7.86 µM vs. 2.31–7.78 µM, L-AABA (Supplementary Table 3). L-BAIBA was only detected in six of 136 younger subjects and in none of 54 older female cohort. When we further stratified these women based on the age: 48–59 years old and 60–80 years old, GABA levels were significanty lower in osteoporotic patients as compared with non-osteoporotic individuals who are 60–80 years old ($p = 0.040$), but no difference for those of 48–59 years old ($p = 0.921$) (Fig. 2a, b). A significant positive correlation between serum D-BAIBA concentration and T-score (hip) was obtained in the older women without any fractures (Pearson's 0.575, $p = 0.0026$, Fig. 2d). Intriguingly, this correlation could not be found in women with fractures (Pearson's −0.088, $p = 0.67$, Fig. 2e), despite a rigorous exclusion criteria for those undergoing any treatment, perhaps pointing to the complexity of fractures or potential effects of OF treatments. Then we further compared serum aminobutyric acid isomer levels with different number and locations of fractures. As compared with the group with no fractures, serum D-BAIBA levels were lower in the population with one or two osteoporotic fractures, but at a similar level in the population with 3–4 fractures (Fig. 2f). We did not detect significant differences for GABA and L-AABA (Fig. 2g, h). Higher D-BAIBA levels can be found in serum from patients with axial fractures, though this increased trend is not statistically significant ($p = 0.0516$, Fig. 2i).

Results using serum from the 136 young women cohort (Supplementary Table 4) showed that the levels of these isomers were 0.58–6.20 µM (D-BAIBA), 0.21–0.74 µM (GABA), and 4.48–79.72 µM (L-AABA). L-BAIBA was only detected in the serum samples from six subjects, with the average concentration of 0.115 ± 0.030 µM (range of 0.076–0.157 µM). All these findings revealed an increasing trend in the young or healthy conditions.

Given the highly significant positive association (Spearman's 0.64, $p = 6.5 \times 10^{-17}$) between hip BMDs and body mass index (BMI) (Supplementary Fig. 2); and other BMI-correlated physical parameters, including body weight and waist circumference in this population, we performed additional analyses on the two subgroups based on BMI: 82 lean women with normal BMI (18.5–24.9 kg m$^{-2}$), and 51 obese women with higher BMI (≥25.0 kg m$^{-2}$). Then the populations without osteoporosis/osteopenia (T-score ≥ −1) and with osteoporosis/osteopenia (T-score < −1) were further grouped to identify metabolites significantly associated with BMD/T-score. Spearman's rank correlation analysis was used to investigate the associations between individual aminobutyric acid isomers and different physical parameters including age, hip BMD/T-score, BMI, current smoking, alcohol intake, physical activity, and dairy consumption. We also performed partial Spearman's rank correlation analysis with all 136 samples as well as adjusting for BMI and BMI-related covariates ($p$-value ≤ 0.05 in Spearman correlation tests) to further investigate the pairwise associations between three aminobutyric acids and each physical parameter controlling for age, BMI, alcohol intake, and/or physical activity. Results were summarized in Tables 3 and 4. Both D-BAIBA and GABA exhibit a significant positive association with physical activity in different populations, suggesting that exercise may increase circulating serum concentrations. L-AABA significantly associated with daily alcohol intake. To our knowledge, this result is the first to link AABA with alcohol intake and is consistent with previous findings linking elevated serum AABA to alcoholic liver injury[1,2]. Even though no association was observed for D-BAIBA with hip BMD in 85 young Caucasian women without osteoporosis/osteopenia, it is interesting that, a positive correlation (Spearman's 0.37, $p = 0.024$) and a negative correlation (Spearman's −0.32, $p = 0.029$) were obtained in 38 lean and 47 obese individuals, respectively (Table 3). This suggests that D-BAIBA associates to adiposity. GABA associated with bone mass in all populations without osteoporosis/osteopenia (Table 3).

**Genetic analyses for aminobutyric acid related genes**. We evaluated 12 genes coding for enzymes and receptors involed with generation or signaling of the aminobutyric acids after analyzing five different published meta-analysis (Supplementary Tables 5

**Table 2 Concentrations of isomeric aminobutyric acids in the pooled or individual human or mouse samples in different matrix.**

| Sample type | Species | Ethnicity | Matrix | Gender | Age | Concentration of isomeric aminobutyric acids (µM) | | | | |
|---|---|---|---|---|---|---|---|---|---|---|
| | | | | | | L-BAIBA | D-BAIBA | GABA | L-AABA | D-AABA |
| Pooled healthy | Human | n/a | Serum | n/a | n/a | N.D. | 0.34 ± 0.02 (5.1) | 0.07 ± 0.01 (3.3) | 4.0 ± 0.3 (8.1) | N.D. |
| | Human | n/a | CSF | n/a | n/a | N.D. | 0.14 ± 0.01 (4.4) | 0.24 ± 0.02 (7.1) | 7.9 ± 0.5 (5.0) | N.D. |
| | Mouse | n/a | Serum | n/a | n/a | 0.05 ± 0.01 (10.1) | N.D. | 1.40 ± 0.07 (5.0) | 51.2 ± 1.5 (3.0) | N.D. |
| | Mouse (CD1) | n/a | Plasma | n/a | n/a | N.D. | N.D. | 0.30 ± 0.01 (1.5) | 4.9 ± 0.3 (5.6) | N.D. |
| Individual | Human | African American | Serum | Male | 30 | N.D. | 1.1 ± 0.1 (5.5) | 0.13 ± 0.02 (11.5) | 9.9 ± 0.7 (7.5) | N.D. |
| | Human | | Serum | | 51 | N.D. | 2.3 ± 0.2 (7.8) | 0.20 ± 0.02 (11.0) | 19.0 ± 1.4 (7.4) | N.D. |
| | Human | Hispanic | Serum | Female | 23 | 0.05 ± 0.01 (8.7) | 0.47 ± 0.04 (8.7) | 0.14 ± 0.01 (10.3) | 14.7 ± 0.8 (5.4) | N.D. |
| | Human | | Serum | | 53 | 0.07 ± 0.01 (9.2) | 1.3 ± 0.1 (5.8) | 0.17 ± 0.02 (9.3) | 13.0 ± 0.2 (1.6) | N.D. |
| | Human | Caucasian | Serum | Female | 25 | 0.06 ± 0.01 (9.0) | 0.92 ± 0.04 (4.8) | 0.18 ± 0.01 (5.3) | 7.2 ± 0.7 (9.0) | N.D. |
| | Human | | Serum | | 61 | 0.04 ± 0.01 (10.7) | 0.91 ± 0.04 (4.8) | 0.13 ± 0.01 (5.4) | 10.8 ± 0.9 (8.0) | N.D. |

Mean ± SD (RSD%), n = 5
CSF cerebrospinal fluid, N.D. not determined

and 6). Three genes, G *protein-coupled receptor 41* (*FFAR3*), *glycine receptor subunit alpha-2* (*GLRA2*), and *glycine receptor subunit alpha-4* (*GLRA4*), were excluded due to lack of genome-wide association studies (GWAS) summary statistics. The results of the gene-based analysis for MRGPRD and DPYD resulted from bioinformatics analyses from other GWAS datasets.

Table 5 presents the results of the analyses of the genes associated with the bone-related traits from five GWAS datasets. We identified seven genes significantly associated with BMD-related traits, among which three genes are essential for BAIBA production and its effects, and the other four genes are needed for GABA generation and effects.

The results of MRGPRD, a functional receptor for BAIBA, from UK biobanks studies UKBB 2017, UKBB 2018, and GEnetic Factors for OSteoporosis (GEFOS) Life Course datasets demonstrated that the gene *MRGPRD* was associated with the estimated BMD (eBMD) by quantitative heel ultrasounds (UKBB 2017: false discovery rate (FDR) = 0.0322, UKBB 2018: FDR = 0.014) and total body BMD (GEFOS Life Course: FDR = 0.027). In addition, *UPB1*, a gene encoding β-ureidopropionase belonging to the CN hydrolase family was found significantly associated with eBMD in the UKBB 2018 (FDR = 0.0272). In addition, we performed gene-based analyses on both pooled cohorts and gender-specific cohorts from the GEFOS2 study showing that *UPB1* significantly associated with femoral neck BMD (FN BMD) and lumbar spine BMD (LS BMD) notably identified among the female cohorts.

Two functional receptors for GABA, GABBR2 (gamma-aminobutyric acid type B receptor subunit 2), and GLRA1 (glycine receptor subunit alpha-1), and an essential enzyme for its generation, GAD1 (glutamate decarboxylase), showed an association with total body BMD among the pooled cohorts (Table 5). In addition, *GLRA1* was also significantly associated with total body BMD among the group with age 60 years or over but not in the younger groups. *GABBR2*, identified from the GEFOS ALLFX study, is the only gene significantly related to fracture (FDR = 0.0108) in the present analysis.

**Correlation analysis for aminobutyric acid related genes.** Since osteoporosis is a metabolic syndrome, where inflammation plays a major role, and given the relevance of peripheral blood monocytes (PBM) to the pathogenesis of aging-related, inflammatory, and degenerative diseases[27–29], we investigated the gene expression profiles for the aminobutyric acids-related genes from PBMs. PBM information was available from 122 out of 136 young women subjects with high or low BMD (Supplementary Table 7)[30]. The full list of measured expression levels of seven aminobutyric acids-related genes shown in Table 5 were summarized in Supplementary Table 8. Expression profiles of aminobutyric acids-related genes that only include *DPYD, UPB1, and GABBR1,* were used to investigate the associations with aminobutyric acids and physical parameters. Four genes, *GABBR2, GAD1, GLRA1, and MRGPRD,* were excluded due to their extremely low expression (essentially absent) in peripheral blood monocyte cells. We further investigated the four low-expressed genes via ENCODE[31] (https://www.encodeproject.org/) and GTEx Portal[32] (https://www.gtexportal.org/home/), and found that *GABBR2, GAD1, GLRA1, and MRGPRD* were consistently low-expressed in monocytes CD14$^+$, osteoblasts, and whole blood cells (Table 6 and Supplementary Fig. 3).

Among 122 subjects that have both transcriptomic and metabolomics datasets, we used both Spearman's and partial Spearman's correlation tests to explore the relationships of the aminobutyric acid related genes with aminobutyric acids and physical parameters, and results were summarized in Table 7. Our results revealed a significant and positive association

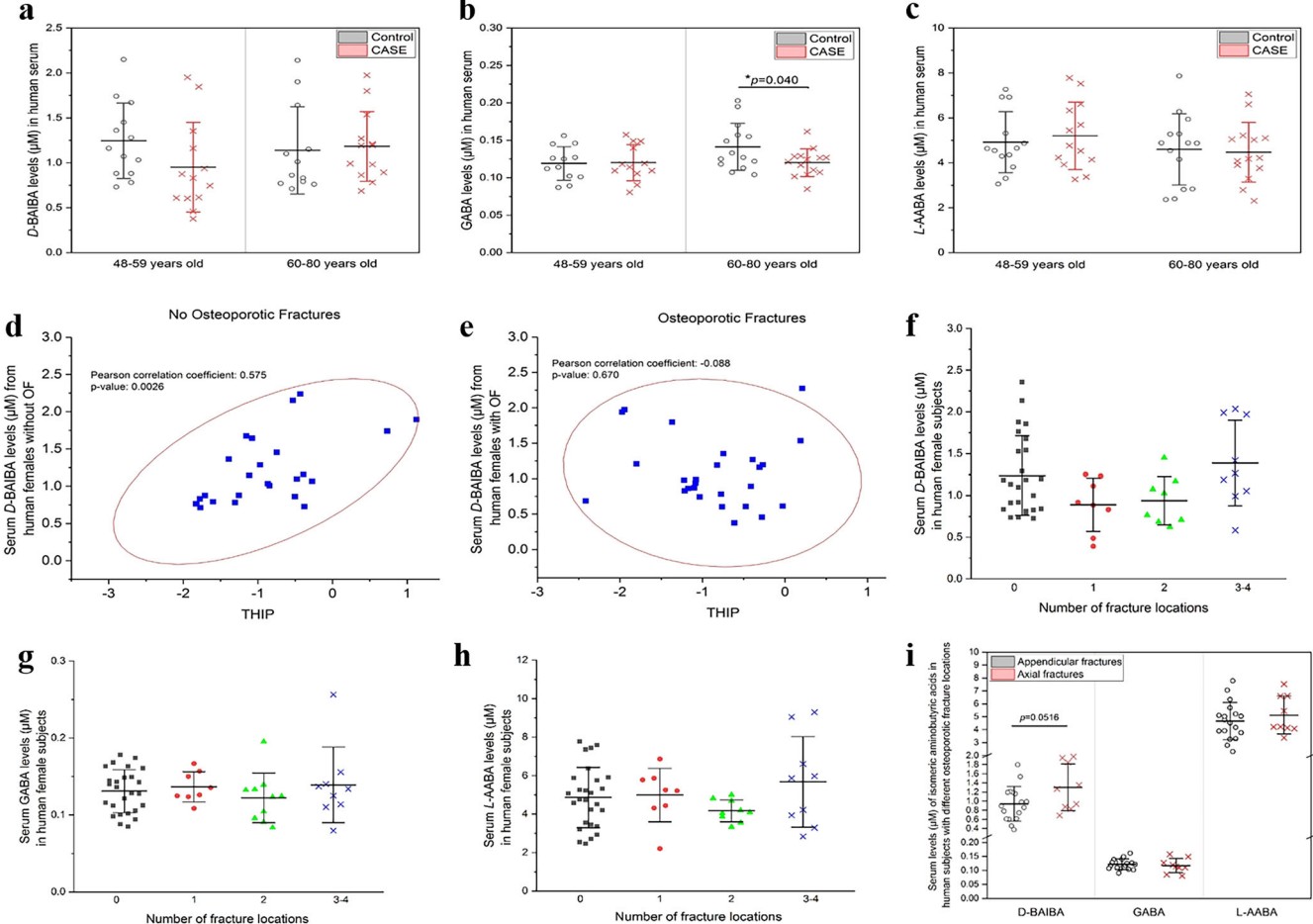

**Fig. 2 Measurement of serum isomeric aminobutyric acid levels in the older women group (48–80 years old) without osteoporotic fractures (Control) or with osteoporotic fractures (CASE). a–c** Comparison of serum levels of isomeric aminobutyric acids, *D*-BAIBA (**a**), GABA (**b**), and *L*-AABA (**c**), in women with (CASE) or without osteoporotic fractures (Control) at different age. Mean ± SD, $n = 13$ pairs in the age of 48–59 years group, and $n = 14$ pairs in the age of 60–80 years group. **d**, **e** Correlations between serum *D*-BAIBA levels and T-score (THIP) from women (48–80 years old), in the group of no osteoporotic fractures (**d**) and osteoporotic fractures (**e**). $n = 27$ for each group. **f–h** Serum levels of aminobutyric acid isomers, **f** D-BAIBA, **g** GABA, and **h** L-AABA, in women with different number of osteoporotic fracture locations. Mean ± SD, $n = 27$ for 0 fracture locations, $n = 8$ for 1 fracture location, $n = 10$ for 2 fracture locations, and $n = 9$ for 3-4 fracture locations. **i** Comparison of serum levels of isomeric aminobutyric acids in patients with appendicular fractures vs. axial fractures. Mean ± SD, $n = 9$ for axial fractures, and $n = 18$ for appendicular fractures.

between two functional enzymes of *D*-BAIBA generation, *DPYD* and *UPB1* ($p < 0.0001$). Moreover, *DPYD* in peripheral blood monocytes exhibited significantly positive association with both physical activity ($\rho' = 0.21$, $p = 0.0188$) and hip BMD ($\rho' = 0.17$, $p = 0.0475$) after controlling influences of physical parameters such as age, BMI, alcohol intake, etc., while *UPB1* level was found to be positively associated with serum *D*-BAIBA level ($\rho = 0.19$, $p = 0.0361$). Additionally, we found a significant correlation between *UPB1* and *L*-AABA ($\rho' = 0.18$, $p = 0.0372$) after controlling age, BMI, etc, and this association has never been reported before. The Spearman correlation test for a GABA receptor coding gene *GABBR1* in transcriptome profiling has established a statistically significant negative correlation to both physical activity ($\rho = -0.17$, $p = 0.0499$) and alcohol intake ($\rho = -0.22$, $p = 0.012$).

## Discussion

Osteoporosis is likely one of the most insidious chronic aging condition, as it commonly goes unnoticed until a fragility fracture occurs. Exercise increases both bone formation and muscle mass

associated with regulation of bone angiogenesis, and has been recommended by the World Health Organization as a non-pharmacologic prophylactic and treatment of osteoporosis[33]. While we have gained a much greater understanding of both osteoporosis and exercise effects at the organismal level, the full cellular-molecular-metabolic machinery is yet unknown. Because of the recent enhanced awareness for the roles of aminobutyric acids in health and disease, their metabolic roles, and their potential involvement with musculoskeletal pathophysiology and exercise, it is essential to quantify precisely these molecules to investigate their specific roles and intracellular signaling mechanisms.

However, as these isomeric analytes are small and polar, they quickly elute out of the column at the void volume, making it difficult to quantify them with baseline seperation under reversed phase HPLC conditions. Previous studies by Rea et al. have demonstrated that a number of co-eluted peaks of biological origin compromised the accurate detection of GABA even with pre-derivatization and an extended retention time of more than 60 min[34]. Incomplete separation of GABA could lead to high basal values of GABA and interactions with unknown compounds that

**Table 3 Association analysis of serum isomeric aminobutyric acids with physical parameters in young Caucasian women (age 21–41 years) using Spearman correlation test.**

| Populations | | Correlations between various serum isomeric aminobutyric acids and physical parameters | | | | | |
|---|---|---|---|---|---|---|---|
| BMI | T-score | Physical parameters | Aminobutyric Acids | Precursors | Sample size | $\rho$ | *p*-value |
| All | All | Physical activity | GABA | Glutamine | 136 | 0.28 | 0.0012 |
| | | Alcohol intake | D-BAIBA | Thymine | 136 | 0.19 | 0.032 |
| | | | L-AABA | Methionine, threonine | 136 | 0.25 | 0.0029 |
| | ≥−1 (No osteopenia or osteoporosis) | BMD | GABA | Glutamine | 85 | −0.25 | 0.021 |
| | | Physical activity | GABA | Glutamine | 85 | 0.34 | 0.0016 |
| | | | D-BAIBA | Thymine | 85 | 0.24 | 0.026 |
| | <−1 (osteoporosis) | Alcohol intake | L-AABA | Methionine, threonine | 51 | 0.38 | 0.006 |
| 18.5–24.9 kg m$^{-2}$ (Lean) | All | Physical activity | GABA | Glutamine | 82 | 0.31 | 0.0055 |
| | | Alcohol intake | L-AABA | Methionine, threonine | 82 | 0.22 | 0.047 |
| | ≥−1 (No osteopenia or osteoporosis) | BMD | D-BAIBA | Thymine | 38 | 0.37 | 0.024 |
| | | Physical activity | GABA | Glutamine | 38 | 0.39 | 0.018 |
| | | | D-BAIBA | Thymine | 38 | 0.33 | 0.044 |
| | <−1 (osteoporosis) | Alcohol intake | L-AABA | Methionine, threonine | 44 | 0.38 | 0.012 |
| ≥25.0 kg m$^{-2}$ (Overweight, obese, extremely obese) | All | Alcohol intake | L-AABA | Methionine, threonine | 51 | 0.41 | 0.0029 |
| | ≥−1 (No osteopenia or osteoporosis) | BMD | D-BAIBA | Thymine | 47 | −0.32 | 0.029 |
| | | Alcohol intake | L-AABA | Methionine, threonine | 47 | 0.40 | 0.0053 |

$\rho$ Spearman's ranked correlation coefficient

**Table 4 Summary of association analysis of serum isomeric aminobutyric acids with physical parameters in young Caucasian women (age 21–41 years) using partial Spearman correlation test.**

| Correlation | Sample size | $\rho'$ | 95% lower bound | 95% upper bound | *p*-value | Control |
|---|---|---|---|---|---|---|
| L-AABA vs. alcohol intake | 136 | 0.27 | 0.1 | 0.42 | 0.0021 | Age, BMI, physical activity |
| D-BAIBA vs. physical activity | 136 | 0.25 | 0.08 | 0.41 | 0.0041 | Age, BMI, alcohol intake |

$\rho'$ partial Spearman's ranked correlation coefficient

mask the true pharmacological efficacy of GABA[34]. In this study, our method achieved baseline separation for these six isomers using our newly developed LC-MS/MS method (Fig. 1a). In addition, to our knowledge, this is the first report of complete, simultaneous separation of these isomers/enantiomers by liquid chromatography, without the time-consuming derivatization process, thereby enhancing sensitivity and efficiency of sample analysis and insuring accurate measurement of each single isomer.

Previous studies have reported reduced serum BAIBA levels of 0.21–0.61 μM in patients with significantly reduced physical activity such as those on hemodialysis[13]. Animal research suggested that BAIBA levels were about 0.6–2 μM in mouse plasma[15]. The normal concentration range for AABA has been reported to be less than 41 μM in human plasma[35], but the concentration could increase up to 151 μM under pathological conditions[35]. Even though these studies did not quantitate separate enantiomers, the LOQ values and linear range of the calibration curves for both R(D) and S(L) enantiomers of BAIBA or AABA in the present LC-MS/MS method seem in excellent agreement with the total levels in control and disease serum. Similarly, the plasma/serum GABA levels in the healthy human subjects typically vary from 0.2 to 1.0 μM[36–38], but under neurological disorders, GABA concentrations decrease to 65–275 nM in CSF and 179–498 nM in plasma[39]. Thus, our method could be suitable for monitoring and screening GABA levels from differential biological samples under healthy or pathological conditions.

To demonstrate the clinical applications of the new methodology, we quantified aminobutyric acid isomers in serum samples from human female populations with and without osteoporosis. In the older women population, significantly lower levels of GABA in the OF group for the subjects of 60–80 years old demonstrated a highly significant association between GABA and bone function during aging (Fig. 2b). Previous studies showed that GABA can stimulate osteoblastogenesis via the activation of GABA B receptor, which is predominantly expressed by osteoblasts[40,41], to favor bone formation. Additionally, GABA may also mitigate osteoporosis by inhibiting the activity of inflammatory cytokines and suppressing oxidative stress-induced cell death[41]. Our findings provide further evidence supporting that GABA might be necessary for optimal bone function particularly in the elderly population (Fig. 3).

A positive correlation between T-score (hip) and D-BAIBA levels was confirmed in the population without OF, suggesting that the individuals with lower BMD have relatively lower D-BAIBA concentrations in serum (Fig. 2d). This finding may provide an explanation for results in Fig. 2a, i.e., serum D-BAIBA levels might decrease during the development of osteoporosis in normal individuals. Interestingly, no similar association was found in the osteoporotic group (Fig. 2e). This discrepancy could

**Table 5 Summary of VEGAS gene-based association results (FDR < 0.05).**

| Aminobutyric acids | Genes | Traits | *p*-value | FDR | Best SNP | SNP *p*-value |
|---|---|---|---|---|---|---|
| BAIBA | *DPYD* | Total body BMD[a] | 0.011 | 0.049 | rs9701777 | 0.00052 |
| | *MRGPRD* | Total body BMD[a] | 0.0067 | 0.049 | rs143309852 | 0.014 |
| | | eBMD[b] | 0.0040 | 0.032 | rs34847539 | 0.00029 |
| | | eBMD[c] | 0.0017 | 0.014 | rs34847539 | 0.000049 |
| | *UPB1* | eBMD[c] | 0.0068 | 0.027 | rs5760459 | 0.00070 |
| | | FN BMD[d] (Female cohort only) | 0.00030 | 0.0015 | rs2070474 | 0.00032 |
| | | LS BMD[d] (Female cohort only) | 0.0026 | 0.021 | rs6004171 | 0.00080 |
| GABA | *GABBR1* | eBMD[c] | 0.000022 | 0.00011 | rs73404750 | $3.5 \times 10^{-12}$ |
| | *GABBR2* | Fracture[e] | 0.0022 | 0.011 | rs11789969 | 0.00017 |
| | | Total body BMD[a] | 0.028 | 0.047 | rs35126377 | 0.00054 |
| | *GAD1* | Total body BMD[a] | 0.023 | 0.047 | rs150390985 | 0.00074 |
| | *GLRA1* | Total body BMD[a] | 0.0043 | 0.022 | rs56177246 | 0.0058 |
| | | Total body BMD[a] (Age ≥ 60 years) | 0.0090 | 0.045 | rs145077031 | 0.0011 |

The *p*-value was calculated by using the VEGAS gene-based association genes coding for enzymes and receptors directly associated with BAIBA and GABA metabolism. False discovery rates with Benjamini-Hochberg multiple test correction was calculated for the adjusted *p*-values and denoted by FDR. Best SNP and SNP *p*-value represent the most significant SNPs within the genes associated with BAIBA and GABA metabolism and the corresponding SNP *p*-value, respectively
*BMD* bone mineral density, *eBMD* bone mineral density estimated from quantitative heel ultrasounds, *FB BMD* femoral neck BMD, *LS-BMD* lumbar spine BMD
[a]GEFOS Life Course
[b]UKBB 2017
[c]UKBB 2018
[d]GEFOS2
[e]GEFOS ALLFX

**Table 6 Expression levels of aminobutyric acid related genes.**

| Gene | Ensembl ID | Measured expression | | Reference expression | | |
|---|---|---|---|---|---|---|
| | | Peripheral blood monocytes | Peripheral blood monocytes | RNA-seq (ENCODE) | | GTEx |
| | | | | Monocytes CD14+ | Osteoblasts | Whole blood |
| | | (Average TPM) | (Median TPM) | (Average RPKM) | (Average RPKM) | (Median TPM) |
| *DPYD* | ENSG00000188641 | 155.9 | 170.7 | 6.7 | 4 | 20.1 |
| *MRGPRD* | ENSG00000172938 | 0 | 0 | 0 | 0 | 0 |
| *UPB1* | ENSG00000100024 | 0.3 | 0.2 | 0 | 0 | 1.2 |
| *GABBR1* | ENSG00000204681 | 39.4 | 31 | 23.6 | 0.2 | 19.6 |
| *GABBR2* | ENSG00000136928 | 0 | 0 | 0 | 0 | 0 |
| *GAD1* | ENSG00000128683 | 0 | 0 | 0 | 0 | 0 |
| *GLRA1* | ENSG00000145888 | 0 | 0 | 0 | 0 | 0 |

the reference expression profiles for other cells/tissue were obtained from ENCODE (https://www.encodeproject.org/) and GTEx Portal (https://www.gtexportal.org/home/) on October 30, 2019
*TPM* transcripts per million, *RPKM* reads per kilobase million

be due to surgical and/or medical treatments after fracture occurance, particularly for those patients with multiple fractures. Such treatments might help stabilize or increase bone formation, leading to some restoration of BAIBA levels.

An intriguing finding was that we only detected *L*-BAIBA in six subjects out of the 190 human subjects studied whereas in mice *L* and not *D*-BAIBA is the major enantiomer in serum. We demonstrated in mice that *L*-BAIBA and not *D*-BAIBA is produced by contracted muscle[18] and it has been shown that total BAIBA levels are elevated after exercise[15]. Therefore, it will be important to perform time course studies using subjects subjected to different forms of exercise to determine if L-BAIBA is also elevated in humans in response to exercise.

It has been reported that peak BMD achieved and maintained in individuals aged 20–40 years is the most powerful predictor of post-menopausal osteoporosis[42]. One of our major goals of focusing these studies on a younger group of women is to attempt to identify biomarkers for early detection of osteopenia/osteoporosis. A positive correlation ($p = 0.024$) was obtained between *D*-BAIBA and BMD in the women with normal BMI, (Table 2) and this result is consistent with what we found in the older

women without osteoporotic fractures: higher serum *D*-BAIBA is linked with higher bone density. A recent study using MC3T3-E1 cells has reported that BAIBA can dramatically stimulate proliferation and differentiation of osteoblasts via activating ROS signaling pathways, inducing bone formation[43]. Interestingly, *D*-BAIBA was found to be inversely associated with BMD ($p = 0.029$) in the populations of overweight, obese, or extremely obese in our study. Gerszten's group has reported that plasma concentration of total BAIBA is negatively associated with metabolic risk factors such as triglycerides, total cholesterol, BMI, etc.[15], suggesting a lower BAIBA level in the obese as compared with lean populations. Moreover, a significant elevation of serum levels of various hormones, cytokines, conventional metabolites, amino acids, and fatty acids occurs in obese compared to lean subjects[44]. All these findings might provide insights for potential apparent discrepancies in observed associations between *D*-BAIBA vs BMD when specific populations present with different metabolic profiles, and further supports its use as a new biomarker for many of these conditions.

To further our understanding of these important associations between BAIBA/GABA and BMD and fractures, we resorted to

**Table 7 Summary significant results of associations between genes vs. aminobutyric acids or physical parameters in 122 young women subjects with high or low BMD.**

| Correlations between genes and aminobutyric acids/physical parameters | Sample size | Spearman correlation analysis | | Partial Spearman correlation analysis | | Control |
|---|---|---|---|---|---|---|
| | | ρ (95% CI) | p-value | ρ' (95% CI) | p-value | |
| DPYD vs. UPB1 | 122 | **0.4 (0.25, 0.54)** | **<0.0001** | **0.4 (0.24, 0.53)** | **<0.0001** | Age, BMI, AI, PA |
| DPYD vs. Hip BMD | 122 | 0.02 (−0.15, 0.19) | 0.8229 | **0.17 (0, 0.34)** | **0.0475** | Age, BMI, AI, PA |
| DPYD vs. physical activity | 122 | −0.13 (−0.29, 0.05) | 0.1558 | **0.21 (0.04, 0.37)** | **0.0188** | Age, BMI, AI |
| GABBR1 vs. alcohol intake | 122 | **−0.22 (−0.38, −0.05)** | **0.012** | 0.09 (−0.09, 0.26) | 0.322 | Age, BMI, PA |
| GABBR1 vs. physical activity | 122 | **−0.17 (−0.34, 0)** | **0.0499** | 0.07 (−0.11, 0.24) | 0.4553 | Age, BMI, AI |
| UPB1 vs. D-BAIBA | 122 | **0.19 (0.01, 0.36)** | **0.0361** | 0.09 (−0.08, 0.26) | 0.2918 | Age, BMI, AI, PA |
| UPB1 vs. L-AABA | 122 | −0.04 (−0.22, 0.14) | 0.6652 | **0.18 (0.01, 0.35)** | **0.0372** | Age, BMI, AI, PA |
| GABBR1 vs. L-AABA | 122 | −0.13 (−0.3, 0.05) | 0.1667 | **0.22 (0.05, 0.38)** | **0.0124** | Age, BMI, AI, PA |

Bold values indicate p-value < 0.05
ρ Spearman's ranked correlation coefficient, ρ' partial Spearman's ranked correlation coefficient, CI confidence interval for ρ/ρ', BMD bone mineral density, AI alcohol intake, PA physical activity

bioinformatics analyses based on five powerful GWAS studies of specific genes associated with BMD-related traits. Strikingly, seven of the 12 genes related to BAIBA/GABA showed significant single nucleotide polymorphisms (SNPs) that are associated with BMD and/or fractures, and all these seven genes are essential for the proper metabolism and effects of BAIBA and GABA. These findings point to the potential genetic modifiers of aminobutyric acids generation/metabolism. Activity/exercise is a major modulator of BAIBA, and we now propose that GABA by controlling muscle tonicity could also exert modulatory effects on levels of BAIBA and its effects (Fig. 3). Our bioinformatics discovered that the SNPs in a key enzyme, UPB1, and a major signaling receptor, MRGPRD, directly involved with availabilty and effects of BAIBA are associated with BMD. This finding is further strongly supported by our transcriptomics analysis of RNA-seq data in peripheral monocytes isolated from the 122 young women cohort. The expression level of the gene *UPB1* in peripheral blood monocytes is positively associated with serum *D*-BAIBA level and the gene *DPYD*, encoding an enzyme necessary for *D*-BAIBA generation, exhibits significantly positive association with both physical activity and hip BMD (Table 7). All these results provide strong rationale or support for the positive association observed between *D*-BAIBA serum levels and hip T-scores in healthy women. This study also suggests for the first time that peripheral blood monocytes may have the capacity to generate *D*-BAIBA, to our knowledge.

Furthermore, we discovered that two genes coding for the GABA receptors *GABBR2* and *GLRA1* and the gene coding for glutamate decarboxylase *GAD1*, showed an association with total body BMD, and remarkably *GABBR2* associated with bone fractures. GABBR2 belongs to the G-protein-coupled receptor family C and GABA-B receptor subfamily[45]. The GABA-B receptors inhibit neuronal activity by regulating the release of neurotransmitters, and the activity of ion channels and adenylyl cyclase[46]. Known diseases associated with this gene include neurodevelopmental disorders[47]. *GAD1* codes for the enzyme responsible for catalyzing the production of gamma-aminobutyric acid from *L*-glutamic acid, and is associated with diseases such as cerebral palsy, spastic quadriplegic, and inherited congenital spastic tetraplegia[48,49]. It may also play a role in stiff-person syndrome[50]. *GLRA1* codes for a subunit of the pentameric inhibitory glycine receptor. Diseases associated with GLRA1 include hyperekplexia, which induce severe stiffness[51]. Another G-protein-coupled receptor family C group 6 subtype A (GPRC6A), though not directly related to BAIBA and GABA, is very important for bone and muscle function as it is activated by amino acids with a preference for the basic and small amino acids such as *L*-Lys, *L*-Arg and *L*-ornithine, and is also directly activated by extracellular calcium and osteocalcin[52,53]. Recently, multiple physiological abnormalities including a disruption in bone metabolism have been reported in *GPRC6A*(−/−) mice[54], and its function in osteoblasts was further reported in humans[55]. These data point to the much broader role of GABA on basic skeletal muscle tone, which influences overall musculoskeletal function and body metabolism. In light of current new concepts on bone-muscle biochemical crosstalk, exploring the functions of BAIBA and GABA as molecules mediating such crosstalk could prove to be very important for the exploration of new therapeutic interventions for musculoskeletal diseases. It is tantalizing to propose that basic skeletal muscle tonus could be way more important as a basic signaling mechanisms than we previously envisioned (Fig. 3).

Exercise increases the release of myokines such as BAIBA, but the exact mechanisms of the beneficial effects of exercise on musculoskeletal health are far from being fully understood. Previous studies have reported elevated BAIBA in plasma after

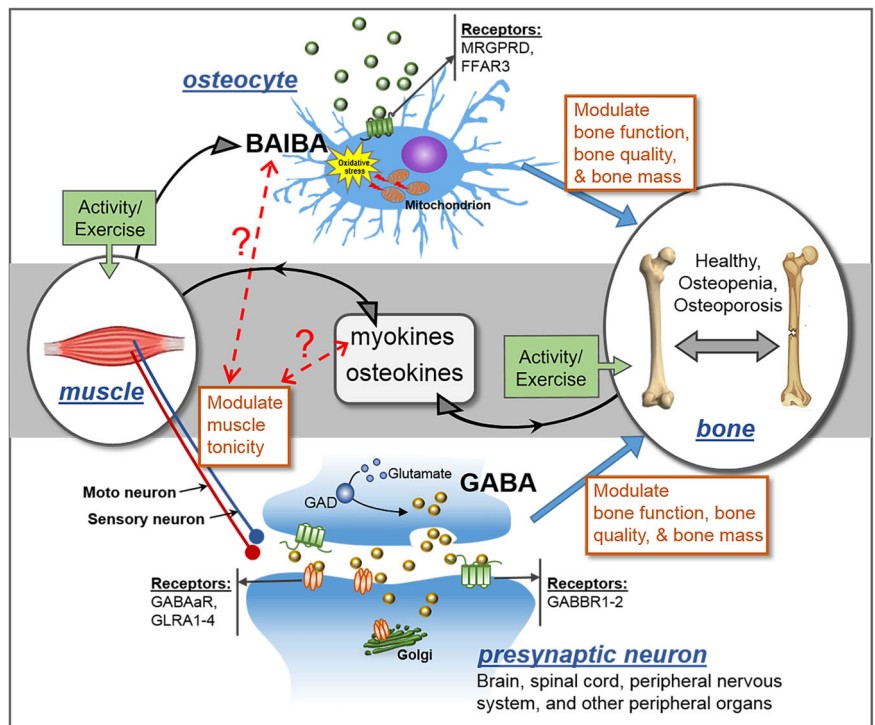

**Fig. 3 Generation and signaling pathways of GABA and BAIBA in bone-muscle crosstalk.** MRGPRD, MAS-related GPR family member D; FFAR3, G-protein-coupled receptor 41; GABAaR, gamma-aminobutyric acid type A receptor; GABBR1-2, gamma-aminobutyric acid type B receptor subunit 1–2; GLRA1-4, glycine receptor alpha 1–4; GAD, glutamate decarboxylase. GABA is generated in CNS, spinal cord, and though its central and peripheral action controls muscle tonicity. BAIBA is a myokine secreted from skeletal muscles with known direct effects in bone/osteocytes in mice. Exercise increases the secretion of both myokines and osteokines that can have autocrine and paracrine effects. GABA is a major neurotransmitter controling muscle tonicity. The dotted lines with question marks are hypothetical concepts that muscle tonicty might influence the level of release of myokines, which in turn can influence the levels of BAIBA, while BAIBA might further modulate tonicity. The receptors for GABA and BAIBA mediate their functions and potential SNPs we report might act as modifiers of these effects. Therefore, we postulate that muscle tonicity could be a new mechanism for the fine-tuning of myokines release and its effects on bone and muscle.

exercise training intervention in both human and mice studies[15], exercise also increases GABA levels in brain[56], gastrocnemius and quadriceps[7], and systemic circulation[7]. Exercise stimulates the generation of these active compounds from muscle, which can further modulate browning of white fat and insulin resistance[15], regulate osteoblastogenesis and osteoclastogenesis[40], prevent loss of bone mass and muscle function[9,18], suppress production of proinflammatory cytokines[57], and play an important role on bone health[58]. In our study, the circulating concentrations of two isomeric aminobutyric acids, *D*-BAIBA and GABA, showed positive associations with physical activity in population of 136 young to middle-aged female individuals with low or high hip BMDs, lean mass, but no association in obese women (Table 2). This suggests a differential metabolomic profile of *D*-BAIBA and GABA in muscle and adipose tissues, or a significantly differential influence of fatty acid metabolism on generation or metabolism of *D*-BAIBA/GABA.

In summary, one of the major challenges in understanding the physiological and pathophysiological roles of isomeric aminobutyric acids is to successfully separate these highly biologically active compounds. Currently, most of the available epidemiological and experimental studies cannot distinguish between isomers/enantiomers of those compounds, particularly *L*-BAIBA and *D*-BAIBA, and therefore have most likely led to misleading or uninterpretable results. To the best of our knowledge, our work is the first to report the complete separation of these isobaric compounds in liquid chromatography without derivatization. This method offers a quick screening with high sensitivity and accuracy for biomarkers of diseases in the plasma, serum, CSF, or

other biological fluid samples from patients, and would significantly benefit and promote the future research associated with these specific isomer/enantiomers in basic research and clinical settings. This newly developed method has now been successfully applied to osteoporosis clinical research studies, and has revealed specific signatures and correlations in women with low and high BMD and in women with one or more OF. In addition, our systematic bioinformatics studies showed that specific SNPs in BAIBA and GABA receptors important for their metabolism and actions are associated with BMD and fractures. These studies open new possibilities for the utilization of aminobutyric acids for the potential diagnosis and prognosis of osteoporosis, and likely a host of musculoskeletal diseases.

## Methods

**Chemicals and reagents.** Pooled healthy human sera and mouse serum were purchased from Sigma–Aldrich (St. Louis, MO). CD1 mouse plasma K2 EDTA was obtained from Innovative Research Inc. (Novi, MI). Pooled normal human cerebrospinal fluid was purchased from Fisher Scientific (Pittsburgh, PA). Human serum from donors with different backgrounds of ethnicity, gender, and age were purchased from Zen-Bio, Inc. (Research Triangle Park, NC). (S)-3-aminoisobutyric acid (*L*-BAIBA) and (R)-3-aminoisobutyric acid (*D*-BAIBA) were purchased from Adipogen Corp. (San Diego, CA), (S)-2-aminobutyric acid (*L*-AABA) and (R)-2-aminobutyric acid (*D*-AABA) were purchased from Thermo Fisher Scientific (Waltham, MA), 4-aminobutyric acid (GABA), (S)-3-aminobutyric acid (*L*-BABA), and 4-aminobutyric-d$_6$ acid (GABA-d$_6$) were purchased from Sigma–Aldrich (St. Louis, MO). DL-2-aminobutyric-2,3,3,4,4,4-d$_6$ acid (*D,L*-AABA-d$_6$) and (±)-3-amino-iso-butyric-2,3,3-d$_3$ acid (*D,L*-BAIBA-d$_3$) were obtained from CDN Isotopes (Pointe-Claire, Quebec, Canada). Monocyte Isolation Kit II was purchased from Miltenyi Biotec Gmbh (Bergisch Gladbach, Germany). AllPrep RNA Universal Kit was purchased from Qiagen (Germantown, MD). KAPA RNA Hyper kit with RiboErase was obtained from KAPA Biosystems (Wilmington, MA). Qubit ds DNA

HS Assay kit was purchased from Life Technologies (Carlsbad, CA). Formic acid (reagent grade, ≥95%), ammonium formate, bovine serum albumin, Histopaque-1077, and ethylenediaminetetraacetic acid (EDTA) were obtained from Sigma–Aldrich (St. Louis, MO). PBS was purchased from Fisher Scientific (Pittsburgh, PA). HPLC-MS grade acetonitrile, water, and methanol were purchased from J.T. Baker (Phillipsburg, NJ).

**LC-MS/MS conditions**. The LC system was equipped with pumps A and B (LC-30AD), and autosampler (SIL-30AC) (Shimadzu Scientific Instruments, Inc., Tokyo, Japan). The LC separation was conducted on a chiral SPP-TeicoShell, 150 × 4.6 mm, 2.7 μm column (AZYP LLC., Arlington, TX) configured with a Max-RP 50 × 2.0 mm as guard column (Phenomenex, Torrance, CA). The analytical column was maintained at room temperature. The mobile-phase consists of methanol (A) and buffer (B). The buffer is containing 0.005% formic acid and 2.5 mM ammonium formate in water. The LC gradient program was: 0.0–10 min, 25% B; 10–10.1 min, 25–98% B; 10.1–17.0 min, 98% B; 17.1–25 min, 25% B. The flow rate was 0.6 mL min⁻¹.

The LC-MS/MS analysis was performed on a Shimadzu LCMS-8050 triple quadrupole mass spectrometer. The instrument was operated and optimized under positive electrospray and multiple reaction monitoring modes (+ESI MRM) using pure standard solutions. The optimized conditions are as follows: interface voltage, 4.0 kV; interface temperature, 300 °C; DL temperature, 300 °C; heating block temperature, 400 °C; drying gas (N₂), 10 L min⁻¹; nebulizing gas (N₂), 3 L min⁻¹; heating gas (Air), 10 L min⁻¹; CID gas (Ar), 230 kPa. The $m/z$ transitions (precursor to product ions) and their tuning voltages were selected based on the best MRM responses from instrumental method optimization software and were presented in Supplementary Table 1. All analyses and data processing were completed on Shimadzu LabSolutions V5.91 software.

The protonated molecular ion ($m/z$ 104⁺, M + H) for all aminobutyric acid isomers produced similar CID fragment ions: $m/z$ 86⁺ ([M + H-H₂O]⁺), $m/z$ 87⁺ ([M + H-NH₃]⁺), $m/z$ 69⁺ ([M + H-H₂O-NH₃]⁺), and $m/z$ 58⁺ ([M + H-HCOOH]⁺). The $m/z$ transitions that can be used for quantification are $m/z$ 104⁺ → 86⁺, $m/z$ 104⁺ → 87⁺, $m/z$ 104⁺ → 69⁺, and $m/z$ 104⁺ → 58⁺. The MRM mass transitions used (Supplementary Table 1) in this method were $m/z$ 104⁺ → 58⁺ for L- and D-AABA, and $m/z$ 104⁺→86⁺ or 69⁺ for the other four isomers based on the optimization results obtained using the Shimadzu automatic method optimization software. After a series of mobile-phase optimization steps with different combinations of organic and aqueous compositions (water, methanol, and acetonitrile) in various amounts of different additives (formic acid, acetic acid, trifluoroacetic acid, ammonium formate, and ammonium acetate), baseline resolution of these six isomers was achieved (Fig. 1a) without a time-consuming derivatization process. The mobile-phase composition consists of (A) methanol and (B) 0.005% formic acid and 2.5 mM ammonium formate in water.

**Sample preparation for evaluation of matrix effect**. Matrix effect is defined as the effect of co-eluting residual matrix components on the ionization of the target analyte, thus matrix can have a significant influence on the accuracy, precision, and robustness of bioanalytical methods, particularly for LC-electrospray ionization (ESI)-MS/MS methods[24]. BSA dissolved in PBS has been widely used in the past to mimic plasma/serum in bioanalysis[59], and was considered the best choice among other methods as it contains the most similar chemical background to most biological fluids[60,61]. In this study, we selected 5% BSA-PBS as a surrogate matrix to establish the standard calibration curves and validate the method. The following parallelism tests were performed to verify the similarity of matrix effects exerted by 5% BSA-PBS and the four kinds of biological fluids including human serum, human CSF, mouse serum and mouse plasma.

Stock solutions of each standard and internal standard (IS) were prepared individually at the concentration of 0.2 M in water. Five isomeric aminobutyric acid standards (L-BAIBA, D-BAIBA, GABA, L-AABA, and D-AABA) of interest and their corresponding deuterated compounds (Supplementary Table 1) were spiked in surrogate matrix 5% BSA-PBS (5% BSA in PBS, pH 7.4, w/v) as well as human serum, human CSF, mouse serum, and mouse plasma. Eleven concentration-level solutions were prepared by diluting the stock solution with corresponding matrices: 0.02, 0.04, 0.08, 0.16, 0.32, 0.64, 1.28, 2.56, 5.12, 10.24, and 20.48 μM for GABA, L-BAIBA and D-BAIBA; 0.16, 0.32, 0.64, 1.28, 2.56, 5.12, 10.24, 20.48, 40.96, 81.92, and 163.84 μM for L-AABA and D-AABA. IS mixture solution (D,L-BAIBA-d₃, D,L-AABA-d₆, and GABA-d₆) was prepared at the concentration of 1.2 μM by dilution of stock solution in methanol containing 0.1% formic acid (v/v). Ten microliter of each sample was mixed with 10 μL IS solution and 35 μL 0.1% formic acid in methanol (v/v). The mixtures were shaken for 20 min at room temperature, and then centrifuged at 12,000 × g, 4 °C for 15 min to precipitate the proteins. The supernatant was transferred to an autosampler vial and 45 μL of each sample was directly injected for LC-MS/MS analysis. Then parallelism of matrix effects was assessed by plotting the concentration-dependent response ratios (standard response normalized by respective IS response) from biological samples to those from 5% BSA-PBS surrogate matrix.

**Preparation of standard calibration curves and QC samples**. As the above evaluations demonstrated that the matrix effects between 5% BSA-PBS and other

four types of samples studied were very similar, the analyte-free 5% BSA-PBS can be used as an ideal surrogate matrix. All samples in the following tests were prepared in 5% BSA-PBS to establish standard calibration curves and validate the method.

The samples of standard calibration curves and QC were prepared by spiking the pure standards in surrogate matrix (5% BSA-PBS). The samples for eight-point calibration curves were prepared by diluting the working solution to the following concentrations: 0.02, 0.04, 0.08, 0.16, 0.64, 2.56, 10.24, and 20.48 μM for L-BAIBA and D-BAIBA; 0.02, 0.04, 0.08, 0.32, 1.28, 5.12, 10.24, and 20.48 μM for GABA; 0.16, 0.32, 0.64, 1.28, 5.12, 20.48, 81.92, and 163.84 μM for L-AABA and D-AABA. QC samples were similarly prepared in quintuplicate at 0.32, 1.28, and 5.12 μM for L-BAIBA and D-BAIBA; 0.16, 0.64, and 2.56 μM for GABA; and 2.56, 10.24, and 40.96 μM for L-AABA and D-AABA. Ten microliters of each sample was mixed with 10 μL IS mixture solution (1.2 μM, in 0.1% formic acid in methanol, v/v) and 35 μL of 0.1% formic acid in methanol (v/v). Then the mixtures were shaken for 20 min at room temperature, followed by the centrifugation at 12,000 × g, 4 °C for 15 min to precipitate protein. Finally, 45 μL of supernatant from each sample was injected for LC-MS/MS analysis.

The limits of detection and quantification (LODs and LOQs, respectively) were determined as the lowest concentration of each analyte detected with a signal-to-noise ratio of 3 and 10, respectively[62,63], and <15% of relative standard deviation (RSD) in five replicates.

The accuracy and precision of the method were determined by analyzing the QC samples in quintuplicate at three concentration levels: 0.32, 1.28, and 5.12 μM for L-BAIBA and D-BAIBA; 0.16, 0.64, and 2.56 μM for GABA; and 2.56, 10.24, and 40.96 μM for L-AABA and D-AABA.

**Preparation of biological fluid samples**. Ten microliter biological fluid samples and same volume of IS mixture solution (1.2 μM, 0.1% formic acid in methanol, v/v) were added to 35 μL of 0.1% formic acid in methanol (v/v), followed by 20 min-shaking at room temperature and another 15 min-centrifugation at 12,000 × g, 4 °C to precipitate protein. The supernatant was directly transferred to an autosampler vial and 45 μL of each sample was injected for LC-MS/MS analysis.

**Peripheral blood monocytes (PBM) isolation**. The human peripheral blood samples were collected from ongoing projects of the Louisiana Osteoporosis Study. First, peripheral blood mononuclear cells (PBMCs) were obtained from 60 milliliter human peripheral blood by density gradient centrifugation on Histopaque-1077. The PBMCs were washed with phosphate-buffered saline (PBS) and then used for PBM isolation with the Monocyte Isolation Kit II according to the manufacture's instructions. The isolated PBMs were examined for purity and counted under microscope using a hemocytometer. The total RNA used for RNA-seq were extracted from the freshly isolated PBMs with the AllPrep RNA Universal Kit and kept at −80 °C for further use.

**Transcriptomics analysis of RNA-seq data**. RNA-seq libraries were constructed from 500 ng RNA of each sample using KAPA RNA Hyper kit with RiboErase following the manufacturer's instructions. Different adaptors were used for multiplexing samples in one sequencing run. Library concentrations and quality were examined by Qubit ds DNA HS Assay kit and Agilent 4150 Tapestation system (Agilent, Santa Clara, CA). The libraries were pooled and diluted to 2 nM in elution buffer (10 mM Tris-HCL, pH 8.5) and then denatured using the Illumina protocol. The denatured libraries were diluted to 10 pM by pre-chilled hybridization buffer and loaded onto Illumina NextSeq 500 (Illumina, San Diego, CA) and run for 75 cycles using a single-read recipe according to the manufacturer's instructions. The data quality examination and demultiplexing procedure were implemented with Illumina SAV and Illumina Bcl2fastq2 version 2.17 program, respectively. TopHat (version 2.0.13)[64] was applied to align sequencing reads to the human reference genome assembly GRCh37 (hg19) for unique alignments. RefSeq transcript annotations were obtained from the UCSC Genome Browser[65], and read fragments aligned to known exons were counted per gene using Htseq (version 0.6.1p1)[66]. All analyses were conducted at the gene level. The RNA-seq raw counts were normalized to the transcripts per million (TPM) unit for the following analyses.

**Recruitment of human subjects**. For the older women group containing both fracturing women and controls, Creighton University Osteoporosis Research Center (ORC) recruited the 54 subjects. The detailed characteristics of the study subjects are shown in Supplementary Table 3. The investigators queried ORC database of persons who had been patients or subjects in previous studies and had consented to be contacted for future studies. ORC purchased lists of primary care physicians and nurse practitioners and sent letters asking them to refer patients. ORC also posted advertisement on the ORC website and their Facebook page. Moreover, ORC contacted orthopedic surgeons to assist in identifying fracturing patients who might qualify for the study. In this cohort, the International Physical Activity Questionnaire was used to quantify physical activity. All of the vertebral fractures were picked up on X-ray; none were acute. Additionally, the following were exclusionary: (1) treatment with Forteo (ever); (2) treatment with Bisphosphonates: Fosamax, Actonel, Boniva, and Reclast. Exception was made if the treatment was more than 10 years prior or only one dose; (3) treatment with bone

active agents within the previous 6 months: Calcitonin, Estrogen, SERM, Depo-Provera, Denosumab, Evista; (4) systemic corticosteroid for more than 6 months duration or any corticoid therapy within the previous 6 months; (5) anticonvulsant therapy within the previous year; and (6) anyone with surgical fixation (metal).

A total of 136 Caucasian females with age between 21–41 years old were recruited from participants in the Louisiana Osteoporosis Study (LOS), a cross-sectional study with ongoing recruitment to build a large sample pool (~20,000 subjects) and database for research studies of osteoporosis and other musculoskeletal diseases/traits[67,68]. The 136 subjects included 65 young women with relatively high BMD and 71 young women with relatively low BMD, corresponding to hip BMD Z scores ≥0.8 and ≤−0.8, respectively. Hip BMD was determined as the combined BMD of the femoral neck, trochanter, and intertrochanteric region measured by Hologic Discovery-A DXA (dual energy X-ray absorptiometry) machine. The DXA machine was calibrated daily, and the coefficient of variation (CV) value of the DXA measurements at total hip on Hologic Discovery-A was 1.0%. For each study subject, weight, height, waist circumference were measured using standard procedures. Lifestyle factors (e.g., exercise, alcohol consumption, smoking, total dairy consumption, etc.) and medical history were assessed by questionnaires. For example, the participant had to first answer whether they performed any physical activity (with a binary answer: yes/no). If the participant answered yes to that question, then the follow-up question asked for the number of days they engaged in physical activity per week. A set of stringent exclusion criteria[69] were adopted to minimize potential confounding effects of non-genetic influence (by physiological and pharmacological conditions) on BMD variation. The detailed characteristics of the study subjects are shown in Supplementary Table 4. Total 122 out of these 136 subjects were involved in transcriptomics analysis (RNA-seq) for their PBMs. Detailed characteristics of these 122 subjects are shown in Supplementary Table 7.

**GWAS analysis for aminobutyric acids-related genes**. We analyzed five different published meta-analysis on genome-wide association studies (GWAS) on BMD of various skeletal sites and fracture traits. The data were downloaded from the GEnetic Factors for OSteoporosis (GEFOS) consortium (http://www.gefos.org/). BMD in these studies is determined by DXA in GEFOS2 and GEFOS Life course studies, whereas the UK biobanks (UKBB) studies (i.e., UKBB2017 and UKBB2018) estimated the BMD by quantitative heel ultrasounds (eBMD). Fracture traits were provided in GEFOS2 ALLFX and the UKBB studies and identified by hospital-based fracture diagnosis (i.e., ICD10 codes) or a self-reported fracture based on questionnaire. Subjects reported any fracture that happened within the past 5 years. Subjects were excluded if they had suffered fracture of the skull, face, hands, and feet or they presented with pathological fractures due to malignancy, atypical femoral fractures, periprosthetic, or healed fracture codes. Supplementary Table 5 is a summary of the GWAS datasets.

In order to evaluate the association of the gene coding for the enzymes and receptors of aminobutyric acids with BMD and fractures, Versatile Gene-based Association Study −2 version 2 (VEGAS2) was implemented. This method is commonly used for GWAS and analysis of summary statistics[70]. The fundamental statistical estimation of VEGAS2 takes into account the linkage disequilibrium (LD) of SNPs within a gene and performs permutation-based simulation to obtain gene-based p-values. We used the association signal from all the SNPs within the region of each gene. The false discovery rate (FDR) was used to address multiple comparisons issue and the adjusted p-values was obtained for the enzymes- and receptors-coding genes for BAIBA and GABA, as shown in Supplementary Table 6. All the GWAS summary data were mapped to hg19 human reference genome downloaded from UCSC genome browser (http://hgdownload.cse.ucsc.edu/goldenPath/hg19/database/). A gene with an adjusted p-value FDR < 0.05 was considered significant.

**Statistics analysis and reproducibility**. Data are presented as mean ± SD of all samples in multiple experiments. Comparison analysis was performed using two-tailed Student's t-test analysis for two-group comparisons, or one-way ANOVA with Tukey's post-Hoc test (α = 0.05) for multiple-group comparisons. Significant outliers in a univariate data set were detected using Grubb's test (α = 0.05) in comparison analysis. Association analysis was performed including: (1) Pearson's correlation for 54 older women with or without osteoporotic fractures, (2) Spearman's rank correlation for 136 young to middle-aged Caucasian women with high or low hip BMDs. Additionaly, partial Spearman's rank correlation, while controlling for age, BMI, alcohol intake, and/or physical activity, was performed in 136 young to middle-aged Caucasian female cohort for all the physical parameters and three aminobutyric acids D-BAIBA, GABA, and L-AABA, by using a package "psych" (version 1.8.12) under R version 3.5.1. p-value ≤ 0.05 was considered as statistically significant difference.

**Reporting summary**. Further information on research design is available in the Nature Research Reporting Summary linked to this article.

**Ethical approval**. Both Creighton University Osteoporosis Research Center and Tulane University Louisiana Osteoporosis Study have all IRB approvals for their clinical studies: Creighton University IRB Committee IRB# 07-14738 for Creighton

ORC studies, and Biomedical IRB Committee IRB# 318016 for Tulane LOS studies. All methods were performed following the relevant policies, regulations and guidelines for analysing datasets from consenting participants and reporting of the findings.

## Data availability
The data that support the findings of this study are available from the authors on reasonable request due to the nature of these data (human subjects); and its associated privacy and HIPPA regulations. See author contributions for specific datasets. All GWAS summary data for the gene-based association tests have been deposited in the GEFOS website (http://www.gefos.org/?q=documents). The gene expression quantification by RNA-Seq for 122 samples will be made available in dbGaP under accession (Accession code phs001960.v1.p1).

## Code availability
The R codes for our analyses are available from the authors on reasonable request, see author contributions for specific datasets.

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

## Acknowledgements

We are grateful for instrumentation support from Shimadzu Scientific Instruments, Inc. This work was supported by NIH-National Institutes of Aging PO1 AG039355 (L.B., M. B.), and R01AG056504 and R01 AG060341 (M.B.), and the George W. and Hazel M. Jay and Evanston Research Endowments (MB). H.S., L.J.Z., K.S., and H.W.D. were partially supported by grants from National Institutes of [R01AR059781, P20GM109036, R01MH107354, R01MH104680, R01GM109068, R01AR069055, U19AG055373, R01DK115679] the Edward G. Schlieder Endowment and the Drs. W.C. Tsai and P.T. Kung Endowment to Tulane University.

## Author contributions

M.B. and Z.W. designed the study. Z.W. and L.B. developed the quantification method and analyzed biological samples. Z.W., L.B., and C.M. analyzed the data and wrote the manuscript with the input from the other authors. M.B., L.F.B., and D.W.A. provided critical advice, discussed the work, and edited the manuscript. H.S., L.J.Z., K.S., and H.D. provided human serum samples and clinical data from 136 young women with low or high BMD, performed RNA-seq assays, gene-based genetic analysis and provided descriptions and summary of the above sample procurement, phenotyping, data analyses, and critical advice/revision. M.K. provided critical advice for LC-MS/MS analysis and edited the manuscript. J.T.L. provided advice for column separation experiments. R.R. and J.L. provided 54 aged osteoporotic women serum samples and critical advice. M.B. directed the entire study.

## Competing interests

The authors declare no competing interests.
