## [Peer Review File · Communications Biology]

Reviewers' comments:

Reviewer #1 (Remarks to the Author):

In this paper authors aimed to investigate the potential role of six underivatized aminobutyric acid isomers (D,L-BAIBA, GABA, D,L-AABA, and L-BABA) in osteoporosis. They performed a validation and genetic study with a concomitant test on several target populations. I completely agree with the authors about identifying biomarkers for early detection of osteopenia/osteoporosis and that the bone-muscle interactions and crosstalk are the new frontier of therapeutic interventions for musculoskeletal diseases.

The paper is well written and I have no concerns regarding the whole methodological and scientific implant of the study.

Minor:

- page 25 line 548: how did the authors quantify physical activity? Do they collect some quantitative variables or just an anamnestic record?
- Page 24/25: do the authors collect informations about eventual surgical or invasive interventions (i.e. vertebroplasty) in the fractured patients?

Reviewer #2 (Remarks to the Author):

Aminobutyric acids and their various derivatives play important roles in cardiovascular and metabolic diseases. Quantification of these non-proteinogenic amino acids has been a challenge. All previous work with these molecules relied on derivatization and complex chemical reactions. This manuscript by the Brotto group fills an important gap in this area of research. It is exciting to see the first method to quickly, accurately, and chromatographically separate and quantify isomeric aminobutyric acids from biological samples, and to apply such information for the field of osteoporosis research.

The authors provide important observation that both GABA and β -aminoisobutyric acid (BAIBA) have positive associations with physical activity in young to middle-aged lean women, and that D-BAIBA was associated with hip BMD in lean individuals without osteoporosis/osteopenia, while lower levels of GABA was observed in older women with osteoporotic fractures. Certainly this manuscript will help to advance the field of aminobutyric acid research applied to musculoskeletal research and aging.

I am enthusiastic about this study and only have minor suggestions to the authors:

- 1) This group last year published a paper in Cell Reports where they showed that BAIBA protected bones and muscles against unloading. It would be good if they could extend their discussion in light of their results to elaborate on how BAIBA and GABA might modulate osteoporosis
- 2) It is unclear if the method is also effective to other biological samples such as urine.
- 3) Do the authors have any information from their 190 subjects on skeletal muscle strength, for example grip?
- 4) Have the authors considered looking at Mediation analyses to determine if some of the variables might work as mediators?

Reviewer #3 (Remarks to the Author):

The manuscript entitled "Quantification of aminobutyric acids and clinical applications for osteoporosis" developed a screening method for aminobutyric acids isoforms and enantiomers yielding correlations with bone mineral density and osteoporotic fracture. In a previous study, they discovered L-BAIBA could promote osteocyte viability and prevent bone and muscle loss. This study expanded the roles of different aminobutyric acids isoforms for osteoporosis and bone-related traits, and found GABA and D-BAIBA might serve as biomarkers for osteoporosis and other aging related musculoskeletal conditions. I think it's a meaningful paper which developed an accurate method to detect aminobutyric acids isoforms and contribute to clinical screening of osteoporosis. Their findings also get insights on genetic mechanisms from these metabolites to BMD and osteoporosis. However, some issues need be addressed are shown as below.

1. It should be cautious to use the words like significant and remarkably except as specific statistics. eg, in Line 194 on Page 10 "remarkably reduced" should mark the p value.
2. It is considered that the change of trend for serum D-BAIBA levels in patients with various fractures was caused by OF treatment, and the levels of GABA and L-AABA seem to be relatively steady regardless of treatment. I think this explanation is premature because the level of GABA and L-AABA have no significant change between populations with and without fracture. Thus, this description needs to be revised.
3. Based on the high correlation between hip BMDs and BMI, I recommend that the hip BMDs should be corrected with BMI and BMI-correlated parameters as covariates rather than division of subgroups.
4. The genetic analyses for aminobutyric acids related genes with VEGAS2 are insufficient to support the hypothetical pathways described in Figure. 3. It is suggested to supplement the correlation analyses between the 7 genes' expression with aminobutyric acid, BMD and BMD-related traits.

Dear Editor,

We enthusiastically submit we our revised manuscript “Quantification of aminobutyric acids and clinical applications for osteoporosis”. We are very much in debt of all reviewers that provided extremely positive guidance for the improvement of our work. We also appreciate that you carefully selected reviewers with expertise in Physical and Rehabilitative Medicine, Ageing biology and regenerative medicine, and skeletal diseases, biomarkers.

We carefully addressed each and all comments by the reviewers.

- 1) We added a substantial amount of new analyses as suggested by all reviewers, including new associations between BAIBA and GABA with specific genes/enzymes controlling their metabolism.
- 2) We discovered new important associations’ even when controlling for co-factors such as age, BMI, alcohol, etc.
- 3) We conducted four different models for the mediation analyses requested by Rev 3.

To accommodate all these requests, we added new supplementary materials in the form of tables and figures.

Below, we provide a detailed and systematic point-by-point response to the reviewers.

Referee expertise:

Referee #1: Physical and Rehabilitative Medicine

Referee #2: Ageing biology and regenerative medicine

Referee #3: Skeletal diseases, biomarkers

Reviewers' comments:

Reviewer #1 (Remarks to the Author):

In this paper authors aimed to investigate the potential role of six underivatized aminobutyric acid isomers (D,L-BAIBA, GABA, D,L-AABA, and L-BABA) in osteoporosis. They performed a validation and genetic study with a concomitant test on several target populations. I completely agree with the authors about identifying biomarkers for early detection of osteopenia/osteoporosis and that the bone-muscle interactions and crosstalk are the new frontier of therapeutic interventions for musculoskeletal diseases.

The paper is well written and I have no concerns regarding the whole methodological and scientific implant of the study.

Minor:

- 1) page 25 line 548: how did the authors quantify physical activity? Do they collect some quantitative variables or just anamnestic record?

Response: We appreciate the enthusiasm of the reviewer for our attempt to identify early biomarkers of osteoporosis. Regarding the minor concern raised, or the older women population containing 54 subjects, Creighton University Osteoporosis Research Center (ORC) used the International Physical Activity Questionnaire. For the young women population containing 136 subjects, Louisiana Osteoporosis Study (LOS) also provided Questionnaire. In this comprehensive questionnaire, the participant had to answer

whether they performed any physical activity (with a binary answer: yes/no). If the participant answered yes to that question, then the follow-up question asked for the number of days they engaged in physical activity per week. For instance, if the participant answered 2, then it suggests that the participant engaged in physical activity 2 days a week. We added this information to the revised methods on page 26-27.

2) Page 24/25: do the authors collect information about eventual surgical or invasive interventions (i.e. vertebroplasty) in the fractured patients?

Response: All of the vertebral fractures were picked up on x-ray; none of them were acute. Thus, any interventions were before the study. Similarly, another criteria for exclusion is anyone with surgical fixation (metal). We added this information to the revised methods on page 26.

Reviewer #2 (Remarks to the Author):

Aminobutyric acids and their various derivatives play important roles in cardiovascular and metabolic diseases. Quantification of these non-proteinogenic amino acids has been a challenge. All previous work with these molecules relied on derivatization and complex chemical reactions. This manuscript by the Brotto group fills an important gap in this area of research. It is exciting to see the first method to quickly, accurately, and chromatographically separate and quantify isomeric aminobutyric acids from biological samples, and to apply such information for the field of osteoporosis research.

The authors provide important observation that both GABA and β -aminoisobutyric acid (BAIBA) have positive associations with physical activity in young to middle-aged lean women, and that D-BAIBA was associated with hip BMD in lean individuals without osteoporosis/osteopenia, while lower levels of GABA was observed in older women with osteoporotic fractures. Certainly this manuscript will help to advance the field of aminobutyric acid research applied to musculoskeletal research and aging.

I am enthusiastic about this study and only have minor suggestions to the authors:

1) This group last year published a paper in Cell Reports where they showed that BAIBA protected bones and muscles against unloading. It would be good if they could extend their discussion in light of their results to elaborate on how BAIBA and GABA might modulate osteoporosis

Response: We very much appreciate this comment and we carefully revised the discussion, details on page 16. We also added a large amount of additional statistical and bioinformatics data, which expanded the depth of our discussion, including newly found associations and mediation analyses.

2) It is unclear if the method is also effective to other biological samples such as urine.

Response: This is an excellent point. LC-MS/MS is susceptible to matrix effects when used to analyze biological samples. Matrix effect is defined as the effect of co-eluting residual matrix components on the ionization of the target analyte, thus matrix can have significant influence on the accuracy, precision, and robustness of bioanalytical methods, particularly for LC-electrospray ionization (ESI)-MS/MS methods

(Matuszewski *et al.*, *Anal Chem.* 1998, 70: 882). In this manuscript, we validated our developed LC-MS/MS method for sensitive and accurate quantification of aminobutyric acids in different biological matrices including serum, plasma and cerebrospinal fluid (CSF) from humans or mice. See additional details in the revised methods page 22-23.

Urine is an aqueous biological samples containing more than 95% water. Unlike serum or plasma but more like CSF, the major interferences in urine proved to be hydrophilic residual components such as inorganic salts. In fact, we have successfully applied this method to quantify aminobutyric acid isomers in urine samples, but we could not conduct these studies in this manuscript because urine was not available for these subjects from Creighton and Tulane.

3) Do the authors have any information from their 190 subjects on skeletal muscle strength, for example grip?

Response: Unfortunately neither Creighton nor Tulane conducted grip strength measurements. We will soon conduct a new series of studies in 120 subjects with such information since we are also interested in skeletal muscle-related traits.

4) Have the authors considered looking at Mediation analyses to determine if some of the variables might work as mediators?

Response: Thanks for this insightful comment. We have, accordingly, done Mediation analyses to explore the potential mediation mechanisms among aminobutyric acid-related genes, aminobutyric acids, hip BMD, and other physical parameters in 122 out of 136 young women cohort who has both transcriptomic and metabolomics data. We considered four causal mediation models to investigate the potential mediator effects of the aminobutyric acids and their corresponding functional enzymes or precursors on hip BMD. Briefly, we implemented two models with the aminobutyric acids as a mediator and aminobutyric acids-related genes as an exposure variable or vice versa. We further used two additional models in which the physical measurements as an exposure variable to investigate the mediation effect of all aminobutyric acids and aminobutyric acids-related genes without and with covariates control, respectively. Mediation analyses shown neither the aminobutyric acid-related genes nor aminobutyric acids significantly mediated effects in these 122 young women subjects. However, we do have additional data to show strong correlations of *D*-BAIBA with physical activity in adjusted models as well as the association of the DPYD (enzyme critical for generation of *D*-BAIBA) with hip BMD. Please see revised manuscript page 10, 12-13, and 25-26, as well as Tables 3, 5-6 on page 37, 39-40, respectively. Collectively, we interpret these findings to endorse the important associations we established between *D*-BAIBA and BMD since it is not other factors that are causing such relationship.

Reviewer #3 (Remarks to the Author):

The manuscript entitled "Quantification of aminobutyric acids and clinical applications for osteoporosis" developed a screening method for aminobutyric acids isoforms and enantiomers yielding correlations with bone mineral density and osteoporotic fracture. In a previous study, they discovered L-BAIBA could promote osteocyte viability and prevent bone and muscle loss. This study expanded the roles of different aminobutyric acids isoforms for osteoporosis and bone-related traits, and found GABA and D-BAIBA might serve as biomarkers for osteoporosis and other aging related musculoskeletal conditions. I think it's a meaningful paper, which developed an accurate method to detect aminobutyric acids isoforms and contribute to clinical screening of osteoporosis. Their findings also get insights on genetic mechanisms from these metabolites to BMD and osteoporosis. However, some issues need be addressed are shown as below.

1. It should be cautious to use the words like significant and remarkably except as specific statistics. eg, in Line 194 on Page 10 “remarkably reduced” should mark the p value.

Response: Yes, we agree with the comments from Reviewers. It has been corrected (page 9 in the revised manuscript).

2. It is considered that the change of trend for serum D-BAIBA levels in patients with various fractures was caused by OF treatment, and the levels of GABA and L-AABA seem to be relatively steady regardless of treatment. I think this explanation is premature because the level of GABA and L-AABA have no significant change between populations with and without fracture. Thus, this description needs to be revised.

Response: We appreciate this comment. We carefully revised the discussion, details on revised manuscript page 16.

3. Based on the high correlation between hip BMDs and BMI, I recommend that the hip BMDs should be corrected with BMI and BMI-correlated parameters as covariates rather than division of subgroups.

Response: Thank you for this suggestion. Accordingly, we have conducted partial Spearman’s rank correlation tests with all the 136 samples as well as adjusting BMI and BMI-related covariates. Partial Spearman’s rank correlation is a rank-based method used to describe the relationship between two variables while removing the effects of several other variables on the relationship.

BMI-related physical parameters, defined as p -value ≤ 0.05 in Spearman correlation tests, are waist circumference, alcohol intake, and physical activity. The Spearman correlation between waist circumference and BMI was strong so that we consider BMI as a proxy adjustment for waist circumference in the following partial Spearman’s rank correlation tests. Besides, we also considered age as a confounder variable due to its statistical correlation with waist circumference, which we also controlled in the analyses. Then we investigated the pairwise associations between three aminobutyric acids and each physical parameter controlling for age, BMI, alcohol intake, and/or physical activity. In summary, we found that the results of partial Spearman’s rank correlation tests were consistent with Spearman’s correlation analyses in subgroups. Significantly positive associations were observed between D-BAIBA vs. physical activity ($p=0.0041$) as well as L-AABA vs. alcohol intake ($p=0.0021$), respectively, in the all-136 young women cohort after excluding the effect of the controlled parameters.

We provided the details about partial Spearman correlation analyses in the revised “Statistical analysis” (page 28-29). We summarized these new results in **Table 3** on page 37. In addition, we explained the rationale for the subgroup analysis and partial Spearman correlation, see details on page 10.

4. The genetic analyses for aminobutyric acids related genes with VEGAS2 are insufficient to support the hypothetical pathways described in **Figure 3**. It is suggested to supplement the correlation analyses between the 7 genes’ expression with aminobutyric acid, BMD and BMD-related traits.

Response: We appreciate the Reviewer’s constructive comment. We have added results and discussions about transcriptomics analysis of RNA-seq data and related correlation analyses (“Results” on page 12-13, “Discussions” on page 18, **Table 5-6** on page 39-40, **Supplementary Fig.3** and **Supplementary Table 9**) for supporting our hypothetical pathways. Detailed information for the methods about peripheral blood monocytes isolation, RNA-seq assay, and statistical analyses; see revised “Methods” (page 25-26, and **Supplementary Table 8**).

In summary, among 122 subjects that have both transcriptomic and metabolomics datasets, we used both Spearman's and partial Spearman's correlation tests to explore the relationships of the aminobutyric acid related genes with aminobutyric acids and physical parameters by comparing gene expression profiles. Firstly, we observed that peripheral lymphocytes have high expression levels of the GABA and D-BAIBA related machinery (**Table 5** on page 39). Furthermore, our results revealed a strong positive association between two functional enzymes of BAIBA generation, *DPYD* and *UPBI* ($p < 0.0001$). Moreover, *DPYD* in peripheral blood monocytes exhibited significantly positive partial Spearman association with both physical activity ($p = 0.0188$) and Hip BMD ($p = 0.0475$) after controlling influences of physical parameters such as age, BMI, Alcohol Intake, etc., while expression level of *UPBI* was found to be positively associated with serum D-BAIBA level ($p = 0.0361$). Additionally, we found significant partial Spearman correlation between *UPBI* and L-AABA ($p = 0.0372$). The Spearman correlation test for a GABA receptor coding gene *GABBR1* in transcriptome profiling has established a statistically significant correlation to both physical activity ($p = 0.0499$) and alcohol intake ($p = 0.012$) (**Table 6** on page 40). All these results further confirm our findings about bioinformatics analyses based on five powerful GWAS studies, and provide a strong support for our hypothetical pathways.

REVIEWERS' COMMENTS:

Reviewer #2 (Remarks to the Author):

The authors have adequately addressed all my comments. Congratulations for such a nice piece of work!

Reviewer #3 (Remarks to the Author):

The authors have addressed all comments and I have no concerns regarding the whole study.